# Neural Networks Trained to Solve Differential Equations Learn General Representations

**Martin Magill**
U. of Ontario Inst. of Tech.
martin.magill1@uoit.net

**Faisal Z. Qureshi**
U. of Ontario Inst. of Tech.
faisal.qureshi@uoit.ca

**Hendrick W. de Haan**
U. of Ontario Inst. of Tech.
hendrick.dehaan@uoit.ca

## Abstract

We introduce a technique based on the singular vector canonical correlation analysis (SVCCA) for measuring the generality of neural network layers across a continuously-parametrized set of tasks. We illustrate this method by studying generality in neural networks trained to solve parametrized boundary value problems based on the Poisson partial differential equation. We find that the first hidden layers are general, and that they learn generalized coordinates over the input domain. Deeper layers are successively more specific. Next, we validate our method against an existing technique that measures layer generality using transfer learning experiments. We find excellent agreement between the two methods, and note that our method is much faster, particularly for continuously-parametrized problems. Finally, we also apply our method to networks trained on MNIST, and show it is consistent with, and complimentary to, another study of intrinsic dimensionality.

## 1  Introduction

Generality of a neural network layer indicates that it can be used successfully in neural networks trained on a variety of tasks [19]. Previously, Yosinski et al. [19] developed a method for measuring layer generality using transfer learning experiments, and used it to compare generality of layers between two image classification tasks. In this work, we will study the generality of layers across a continuously-parametrized set of tasks: a group of similar problems whose details are changed by varying a real number. We found the transfer learning method for measuring generality prohibitively expensive for this task. Instead, by relating generality to similarity, we develop a computationally efficient measure of generality that uses the singular vector canonical correlation analysis (SVCCA).

We demonstrate this method by measuring layer generality in neural networks trained to solve differential equations. We train fully-connected tanh neural networks (NNs) to solve Poisson's equation with a parametrized source term. The parameter of the source defines a family of related boundary value problems (BVPs), and we measure the generality of layers in the trained NNs as the parameter varies. We find the first layers to be general, and deeper layers to be progressively more specific. Using the SVCCA, we are also able to visualize and interpret these general first layers.

We validate our approach by reproducing a subset of our results using the transfer learning experimental protocol of Yosinski et al. [19]. These very different methods produce consistent measurements of generality. Further, our technique is several orders of magnitude faster to compute. Finally, we apply our method to ReLU networks trained on the MNIST dataset [9], and compare to work by Li et al. [11]. We discuss how the two analyses differ, but confirm that our results are consistent with theirs.

The main contributions of this work are:

1. We develop a method for efficiently computing layer generality over a continuously-parametrized family of tasks using the SVCCA.

2. Using this method, we demonstrate generality in the first layers of NNs trained to solve problems from a parametrized family of BVPs. We find that deeper layers become successively more specific to the problem parameter, and that network width can play an important role in determining layer generality.

3. We visualize the principal components of the first layers that were found to be general. We interpret them as generalized coordinates that reflect important subregions of the unit square.

4. We validate our method for measuring layer generality using the transfer learning experimental protocol developed by Yosinski et al. [19]. We find that both approaches identify the same trends in layer generality as network width is varied, but that our approach is significantly more computationally efficient, especially for continuously parametrized tasks.

5. We define a measure of the intrinsic dimensionality of a layer, and contrast it with that of Li et al. [11]. We show the two are consistent for networks trained on the MNIST dataset.

## 1.1 Neural networks for differential equations

The idea to solve differential equations using neural networks was first proposed by Dissanayake and Phan-Thien [3]. They trained neural networks to minimize the loss function

$$\mathcal{L} = \int_\Omega \|G[u](x)\|^2 dV + \int_{\partial\Omega} \|B[u](x)\|^2 dS, \tag{1}$$

where $G$ and $B$ are differential operators on the domain $\Omega$ and its boundary $\partial\Omega$ respectively, $G[u] = 0$ is the differential equation, and $B[u] = 0$ describes boundary conditions. Training data consisted of coordinates $x \in \Omega$ sampled from a mesh, used to numerically approximate the integrals in $\mathcal{L}$ at each epoch. Similar methods were proposed by van Milligen et al. [17] and Lagaris et al. [7]. Many innovations have been made since, most of which were reviewed by Schmidhuber [15] and in a book by Yadav et al. [18]. Sirignano and Spiliopoulos [16] as well as Berg and Nyström [2] illustrated that the training points can be obtained by randomly sampling the domain rather than using a mesh, which significantly enhances performance in higher-dimensional problems. In fact, Sirignano and Spiliopoulos [16] and Han et al. [5] have demonstrated that neural networks can be used to solve partial differential equations in hundreds of dimensions, which is a revolutionary result. Traditionally, such problems have often been considered infeasible, since traditional mesh-based solvers suffer from an exponential growth in computational complexity with increasing problem dimensionality.

There are at least two good reasons for studying neural networks that solve differential equations (referred to hereafter as DENNs). The first is their unique advantages over traditional methods for solving differential equations [2–5, 7, 16, 17]. The second is that they offer an opportunity to study the behaviour of neural networks in a well-understood context [2]. Most applications of neural networks, such as machine vision and natural language processing, involve solving problems that are ill-defined or have no known solutions. Conversely, there exists an enormous body of literature on differential equation problems, detailing when solutions exist, when they are unique, and how they will behave. Indeed, in some cases the exact solutions to the problem can be obtained analytically.

## 1.2 Studying the generality of features with transfer learning

Transfer learning is a major topic in machine learning, reviewed for instance by Pan and Yang [13]. Generally, transfer learning in neural networks entails initializing a recipient neural network using some of the weights from a donor neural network that was previously trained on a related task.

Yosinski et al. [19] developed an experimental protocol for quantifying the generality of neural network layers using transfer learning experiments. They defined generality as the extent to which a layer from a network trained on some task A can be used for another task B. For instance, the first layers of CNNs trained on image data are known to be general: they always converge to the same features, namely Gabor filters (which detect edges) and color blobs (which detect colors) [6, 8, 10].

In the protocol developed by Yosinski et al. [19], the first $n$ layers from a donor network trained on task A are used to initialize the first $n$ layers of a recipient network. The remaining layers of the recipient are randomly initialized, and it is trained on task B. However, the transferred layers are frozen: they are not updated during training on task B. The recipient is expected to perform as well on task B as did the donor on task A if and only if the transferred layers are general.

In practice, however, various other factors can impact the performance of the recipient on task B, so Yosinski et al. [19] also included three control tests. The first control is identical to the actual test, except that the recipient is trained on the original task A; this control identifies any fragile co-adaptation between consecutive layers [19]. The other two controls entail repeating the actual test and the first control, but allowing the transferred layers to be retrained. When the recipient is trained on task A with retraining, performance should return to that of the donor network. When it is trained on task B with retraining, Yosinski et al. [19] found the recipient actually outperformed the donor.

Yosinski et al. [19] successfully used their method to confirm the generality of the first layers of image-based CNNs. Further, they also discovered a previously unknown generality in their second layers. This methodology, however, was constructed for the binary comparison of two tasks A and B. In the present work, we are interested in studying layer generality across a continuously parametrized set of tasks (given by a family of BVPs), and the transfer learning methodology is prohibitively computationally expensive. Instead, we will use a different approach, based on the SVCCA, which we will then validate against the method of Yosinski et al. [19] on a set of test cases.

### 1.3 SVCCA: Singular Vector Canonical Correlation Analysis

Yosinski et al. [19] defined generality of a layer to mean that it can be used successfully in networks performing a variety of tasks. This definition was motivated, however, by observing that the first layers of image-based CNNs converged upon similar features across many network architectures and applications. We argue that these two concepts are related: if a certain representation leads to good performance across a variety of tasks, then well-trained networks learning any of those tasks will discover similar representations. In this spirit, we define a layer to be general across some group of tasks if similar layers are consistently learned by networks trained on any of those tasks. To use this definition to measure generality, then, we require a quantitative measure of layer similarity.

Recently, the SVCCA was demonstrated by Raghu et al. [14] to be a powerful method for measuring the similarity of neural network layers [14]. The SVCCA considers the activation functions of a layer's neurons evaluated at points sampled throughout the network's input domain. In this way it incorporates problem-specific information, and as a result it outperforms older metrics of layer similarity that only consider the weights and biases of a layer. For instance, Li et al. [12] proposed measuring layer similarity by finding neuron permutations that maximized correlation between networks. As a linear algebraic algorithm, however, the SVCCA is more computationally efficient than permutation-based methods. Similarly, Berg and Nyström [2] have concurrently attempted to study the structure of DENNs by analyzing weights and biases directly, but found the results to be too sensitive to the local minima into which their networks converged.

Following Raghu et al. [14], we will use the SVCCA to define a scalar measure of the similarity of two layers. The SVCCA returns canonical directions in which two layers are maximally correlated. They defined the SVCCA similarity $\rho$ of two layers as the average of these optimal correlation values. However, this quantity depends explicitly on the layers' widths, independently of the functions the layers represent. Here, instead of the mean, we will define $\rho$ as the sum of these correlations. Since we typically found that the majority of the correlations were nearly 1.0 or nearly 0.0, this SVCCA similarity roughly measures the number of significant dimensions shared by two layers. In particular, since the SVCCA between a layer and itself is equivalent to a principal component analysis, we will use the SVCCA self-similarity as an approximate measure of a layer's intrinsic dimensionality.

This concept of intrinsic dimensionality differs from that recently proposed by Li et al. [11]. They constrained network weights during training to a random $d$-dimensional subspace for various values of $d$, and defined the intrinsic dimensionality of a given network on a given task as the smallest $d$ for which good performance is achieved. This metric differs from ours in two important ways. First, their algorithm finds the *smallest representation* required to solve a problem, whereas our definition directly analyses the *actual representations* learned in practice. Specifically, they consider a strongly regularized auxilliary problem, whereas we examine given solutions directly. Second, their measure is based on the *performance* of representations, whereas ours measures *structure*. Indeed, since Raghu et al. [14] were able to compress models with little loss of performance by keeping only the first few important SVCCA directions, the remaining important SVCCA directions must describe structures present in layers' representations that do not directly influence network performance. Our method is complimentary to that of Li et al. [11]: theirs finds compact solutions that perform well, which is of practical value, but ours can examine any given network without altering its properties.

## 2 Methodology

### 2.1 Problem definition

Following concurrent work by Berg and Nyström [2], we will study the structure of DENNs on a parametrized family of PDEs. Berg and Nyström [2] used a family of Poisson equations on a deformable domain. They attempted to characterize the properties of the DENN solutions by studying the variances of their weights and biases. However, they reported that their metrics of study were too sensitive to the local minima into which their solutions converged for them to draw conclusions [2].

In this work, we have repeated this experiment, but using the SVCCA as a more robust tool for studying the structure of the solutions. The family of PDEs considered here was

$$\nabla^2 u(x, y) = s(x, y) \quad \text{for} \quad (x, y) \in \Omega, \tag{2}$$
$$u(x, y) = 0, \quad \text{for} \quad (x, y) \in \partial\Omega, \tag{3}$$

where $\Omega = [-1, 1] \times [-1, 1]$ is the domain and $-s(x, y)$ is a nascent delta function given by

$$s(x, y) = -\delta_r(x, y; x', y') = -\frac{\exp\left(-\frac{(x-x')^2+(y-y')^2}{2r^2}\right)}{2\pi r^2}, \tag{4}$$

which satisfies $\lim_{r\to 0} \delta_r(x, y; x', y') = \delta(x - x')\delta(y - y')$, where $\delta$ is the Dirac delta function. For the present work, we will fix $y' = 0$ and $r = 0.1$, and vary only $x'$. Thus, the BVPs describe the electric potential produced by a localized charge distribution on a square domain with grounded edges. The problems are parametrized by $x'$. We relegate deformable domains to future work.

### 2.2 Implementation details

The networks used in this work were all fully-connected with 4 hidden layers of equal width, implemented in TensorFlow [1]. Activation functions were tanh, except in Section 3.4, where ReLU was used. Given inputs $x$ and $y$, the network was trained to directly approximate $u(x, y)$, the solution to a BVP from the family of BVPs described above. Training followed the DGM methodology of Sirignano and Spiliopoulos [16]. More implementation details are discussed in the supplemental material. Since this work was not focused on optimization of performance, we used relatively generic hyperparameters whenever possible to ensure that our results are reasonably general.

## 3 Results

### 3.1 Quantifying layer generality in DENNs using SVCCA

In this section, we use the SVCCA to study the generality of layers in DENNs trained to solve our family of BVPs. We train DENNs to solve the BVPs for a range of $x'$ values, each from four different random initializations per $x'$ value. We will refer to the different random initializations as the first through fourth random seeds for each $x'$ value (see the supplemental material for details about the random seed construction). First, we present results for networks of width 20. We condense our analysis into three metrics, and then study how those metrics vary with network width.

Figure 1 shows the SVCCA similarities computed between the first, third, and fourth hidden layers of networks of width 20. The matrix for the second hidden layer is omitted, but closely resembles that for the first hidden layer. The $(i, j)$th element of the matrices show the SVCCA similarity computed between the given layers of the $i$th and $j$th networks in our dataset. Since the SVCCA similarity does not depend on the order in which the layers are compared, the matrices are symmetric. The black grid lines of the matrices separate layers by the $x'$ values on which they were trained, and the four seeds for each $x'$ are grouped between the black grid lines.

The matrices evidently exhibit a lot of symmetry, and can be decomposed into subregions. The first is the diagonal of the matrices, which contains the self-similarities of the layers, denoted $\rho_{\text{self}}^l$ in the $l$th layer. The second region contains the matrix elements that lie inside the block diagonal formed by the black grid lines, but that are off the main diagonal. These indicate the similarities between layers trained on the same $x'$ values, but from different random seeds, and will be denoted $\rho_{\Delta x'=0}^l$. The remaining matrix elements were found to be equivalent along the block-off-diagonals. These

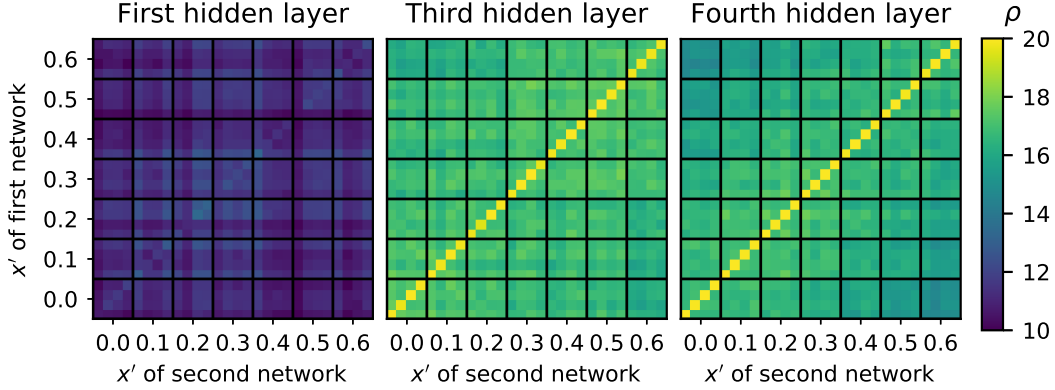

Figure 1: Matrices of layer-wise SVCCA similarities between the first, third, and fourth hidden layers of networks of width 20 trained at various $x'$ values, with four random seeds per position. The black lines group layers on each axis by the $x'$ values at which they were trained. For each $x'$ value, the four entries correspond to four distinct random seeds. Thus the matrix diagonals contain self-similarities, the block diagonals formed by black lines contain similarities across random seeds at a fixed $x'$, and the remaining entries correspond to comparisons between distinct $x'$ values.

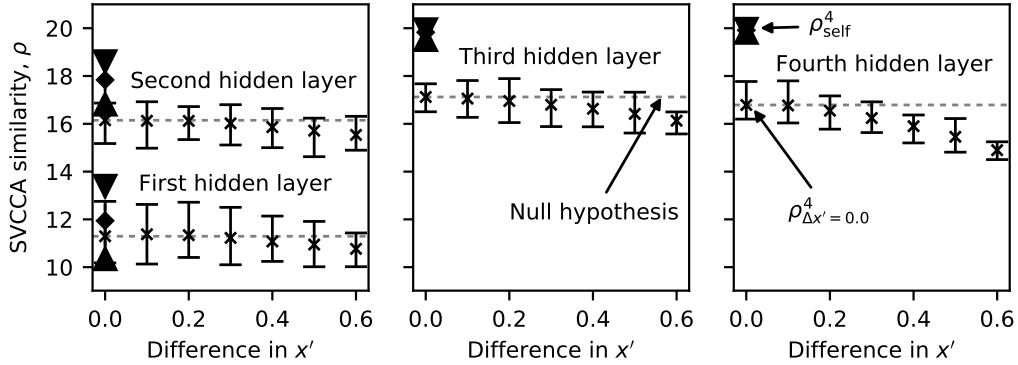

Figure 2: For each layer, crosses show mean similarities between distinct layers as a function of the difference in the $x'$ values at which they were trained. Diamonds show mean self-similarities. For both, error bars indicate maximum and minimum values. The gray lines show the null hypothesis described in the text, namely that the representations are independent of $x'$.

correspond to all similarities computed between $l$th layers from networks trained on $x'$ values that differ by $\Delta x'$, which we will denote $\rho^l_{\Delta x'}$.

With this decomposition in mind, the matrices can be represented more succinctly as the plots shown in Figure 2. The diamonds and their error bars show the mean, minima, and maxima of $\rho^l_{\text{self}}$ in each layer $l$. The crosses and their error bars show the means, minima, and maxima of $\rho^l_{\Delta x'}$ for varying source-to-source distances $\Delta x'$. As described above, the statistics of $\rho^l_{\Delta x'=0}$ were computed excluding the self-similarities $\rho^l_{\text{self}}$. The dashed gray lines show $\langle \rho^l_{\Delta x'=0} \rangle$ for each layer, and are used below to quantify specificity. We show the minima and maxima of the data in order to emphasize that our decomposition of the matrices in Figure 1 accurately reflects the structure of the data.

In the plots of Figure 2, the gap between $\rho^l_{\text{self}}$ and $\rho^l_{\Delta x'=0}$ indicates the extent to which different random initializations trained on the same value of $x'$ converge to the same representations. For this reason, we define the ratio $\langle \rho^l_{\Delta x'=0} \rangle / \langle \rho^l_{\text{self}} \rangle$ as the reproducibility. It measures what fraction of a layer's intrinsic dimensionality is consistently reproduced across different random seeds. We see that, for networks of width 20, the first layer is highly reproducible, and the second is mostly

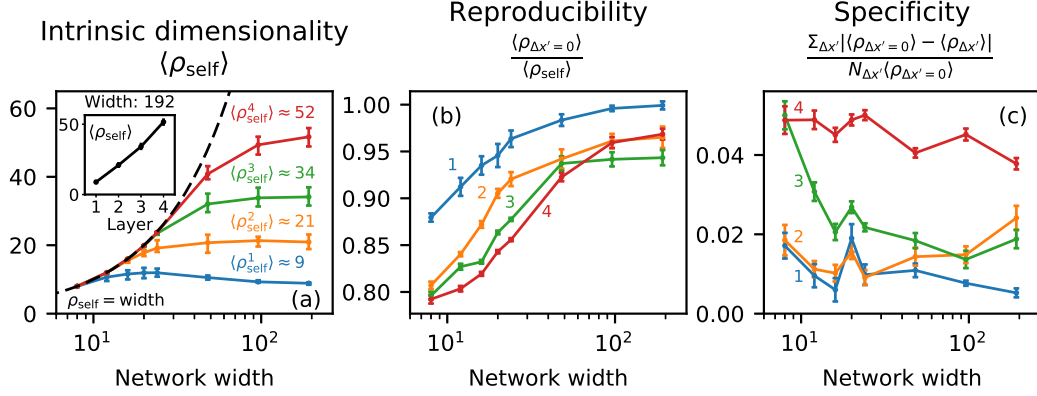

Figure 3: The intrinsic dimensionality, reproducibility, and specificity of the four layers at varying width. The lines indicate mean values. The error bars on intrinsic dimensionality indicate maxima and minima, whereas the error bars on reproducibility and specificity indicate estimated uncertainty on the means (discussed in the supplemental material). Numbers indicate layer numbers. The inset in (a) shows the limiting dimensionalities of the four layers at width 192.

reproducible. Conversely, the third and fourth layers in Figure 2 have a gap of roughly 3 out of 20 between $\langle \rho^l_{\Delta x'=0} \rangle$ and $\langle \rho^l_{\text{self}} \rangle$: networks from different random seeds at the same $x'$ value are consistently dissimilar in about 15% of their canonical components.

We can use the plots of Figure 2 to quantify the generality of the layers. When a layer is general across $x'$, the similarity between layers should not depend on the $x'$ values at which they were trained. Thus the $\rho^l_{\Delta x'}$ values should be distributed no differently than $\rho^l_{\Delta x'=0}$. Visually, when a layer is general, the crosses in Figure 2 should be within error of the dashed grey lines. Similarly, the distance between the crosses and the dashed line is proportional to the specificity of a layer. Thus we can see in Figure 2 that, for networks of width 20, the first and second layers appear to be general, whereas the third and fourth are progressively more specific.

To quantify this, we will define a layer's specificity as the average over $\Delta x'$ of $\left| \langle \rho^l_{\Delta x'=0} \rangle - \langle \rho^l_{\Delta x'} \rangle \right| / \langle \rho^l_{\Delta x'=0} \rangle$. In Figure 2, this is equivalent to the mean distance from the crosses to the dashed grey line, normalized by the height of the dashed grey line. Equivalently, it is the ratio of the area delimited by the crosses and the dashed line to the area under the dashed line. It can also be interpreted as a numerical estimation of the normalized $L_1$ norm of the difference between the measured $\langle \rho^l_{\Delta x'=X} \rangle$ and the null hypothesis of a perfectly general layer. By this definition, a layer will have a specificity of 0 if and only if it has similar representations across all values of $\Delta x'$. Furthermore, the specificity is proportional to how much $\langle \rho^l_{\Delta x'} \rangle$ varies with $\Delta x'$. Thus the specificity metric we defined here is indeed consistent with the accepted definitions of generality and specificity.

The same experiments described above for networks of width 20 were repeated for widths of 8, 12, 16, 24, 48, 96, and 192. Figure 3 shows the measured intrinsic dimensionalities, reproducibilities, and specificities of the four layers. The error bars on the intrinsic dimensionalities show minima and maxima, emphasizing that these measurements were consistent across different values of $x'$ and different random seeds. The error bars on the reproducibility and specificity show the estimated uncertainty on the means, as discussed in the supplemental material.

In narrow networks, the layers' intrinsic dimensionalities (Fig. 3(a)) equal the network width. As the network width increases, these dimensionalities drop below the width, and appear to converge to finite values. We suggest that, for a fixed $x'$ value, there are finite-dimensional representations to which the layers will consistently converge, so long as the networks are wide enough to support those representations. If the networks are too narrow, they converge to some smaller-dimensional projections of those representations. The reproducibility plots (Fig. 3(b)) support this interpretation, as the reproducibilities grow with network width. Furthermore, they are smaller for deeper layers, except in very wide networks where the fourth layer becomes more reproducible than the second and third. This could be related to convergence issues in very wide networks, as discussed below.

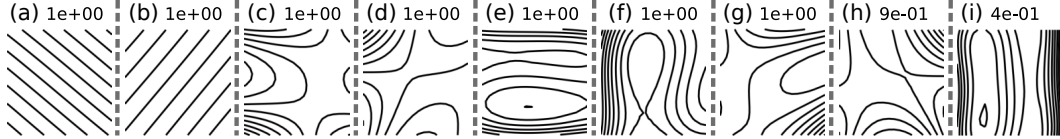

Figure 4: Plots of the first nine principal components of the first layer of a network trained on $x' = 0.6$ (obtained by self-SVCCA). The numbers show the SVCCA correlations of each component.

The limiting dimensionalities increase nearly linearly with layer depth, as shown in the inset of Figure 3(a). This has implications for sparsification of DENNs, as Raghu et al. [14] showed successful sparsification by eliminating low-correlation components of the SVCCA. Similarly, DENN architectures that widen with depth may be more optimal than fixed-width architectures.

The specificity (Fig. 3(c)) varies more richly with network width. Overall, the first layer is most general, and successive layers are progressively more specific. Over small to medium widths, the second layer is nearly as general as the first layer; the third layer transitions from highly specific to quite general; and the fourth layer remains consistently specific. In very wide networks, however, the second and third layers appear to become more specific, whereas the fourth layer becomes somewhat more general. Future work should explore the behaviour at large widths, but we speculate that it may be related to changes in training dynamics at large widths. As discussed in the supplemental material, very wide networks seemed to experience very broad minima in the loss landscape, so our training protocol may have terminated before the layers converged to optimal and general representations.

The overall trends in specificity discussed above are interrupted near widths of 16, 20, and 24. All four layers appear somewhat more general than expected at width 16, and then more specific than expected at width 20. By width 24 and above they resume a more gradual variation with width. This is a surprising result that future work should explore more carefully. It occurs as the network width exceeds the limiting dimensionality of the second layer, which may play a role in this phenomenon.

## 3.2 Visualizing and interpreting the canonical directions

We have shown that the first layers of the DENNs studied here converge to general 9-dimensional representations independent of the parameter $x'$. Figure 4 shows a visualization of the first 9 principal components (obtained by self-SVCCA) of the first layer of a network of width 192 trained at $x' = 0.6$, shown as contour maps. We interpret these as generalized coordinates. The contours are densest where the corresponding coordinates are most sensitive. It is clear that the first 2 of these 9 components capture precisely the same information as $x$ and $y$, but rotated. The remaining components act together to identify 9 regions of interest in the domain: the 4 corners, the 4 walls, and the center. For instance, component (e) describes the distance from the top and bottom walls; component (i) does the same for the left and right walls; and component (d) describes distance to the upper-left corner. We found the first layers could be interpreted this way at any $x'$ and whether we found components by self-SVCCA or cross-SVCCA, and have included examples of this in the supplemental material. The components are always some linear combination of $x$, $y$, and the 9 regions described above.

Surprisingly, we found that the SVCCA was numerically unstable. Repeated analyses of the same networks produced slightly different components, although the correlation vectors were very stable. We see two factors contributing to this problem. Firstly, the first 7 or 8 correlation values of the first layer are all extremely close to 1 and, therefore, to one another. Thus the task of sorting the corresponding components is inevitably ill-conditioned. Second, the components appear to be paired into subspaces, such as the first two in Figure 4. Thus the task of splitting these subspaces into one-dimensional components is also ill-conditioned. We propose that future work should explore component analyses that search for closely-coupled components. This could resolve the numerical stability while also extracting even more structure about the layer representations.

## 3.3 Confirming generality by transfer learning experiments

In this section, we validate the method used to measure generality in Section 3.1 by repeating a subset of our measurements using the transfer learning technique established by Yosinski et al. [19]. We restricted our validation to a subset of cases because the transfer learning technique is significantly

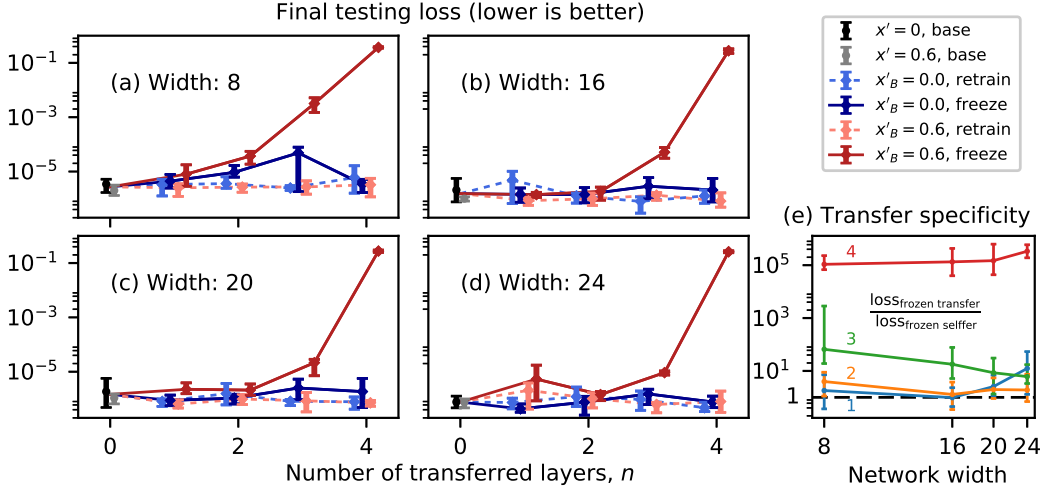

Figure 5: (a-d): Results of the transfer learning experiments conducted on networks of four different widths. Markers indicate means, and error bars indicated maxima and minima. At $n = 0$, the lines pass through the average of the two base cases. Donors were trained at $x'_A = 0$, and recipients at $x'_B$. (e): Measured transfer specificity as a function of network width. Numbers indicate layer number. Markers show the ratio of the mean losses, and error bars show the maximum and minimum ratios over all 16 combinations of the two losses. The dashed line shows a transfer specificity of 1.

more computationally expensive. To this end, we only trained donor networks at $x'_A = 0$ and measured generality towards $x'_B = 0.6$. Following Yosinski et al. [19], we will call the control cases with $x'_B = 0$ the selffer cases, and the experimental cases with $x'_B = 0.6$ the transfer cases. We show the results for widths of 8, 16, 20, and 24 in Figure 5(a-d). Throughout this section, we will refer to the measure of layer specificity we defined in Section 3.1 as the SVCCA specificity, to distinguish it from the measure of layer specificity obtained from the transfer learning experiments, which we call the transfer specificity. In Figure 5(a-d), the transfer specificity is given by the difference between the losses of the frozen transfer group (solid, dark red points) and those of the frozen selffer group (solid, dark blue points). It is immediately clear that the third and fourth layers are much more specific than the first and second at all widths. The specificities of the first, second, and fourth layers do not change very much with width, whereas the third layer appears to become more general with increasing width. These results are in agreement with those found with the SVCCA specificity.

To quantify these differences, we define a transfer specificity metric given by the ratio of the losses between the two frozen groups. This is shown in Figure 5(e), and it can be compared to the SVCCA specificities for the same widths, which lie in the leftmost third of Figure 3(c). The dashed line in Figure 5(e) shows a transfer specificity of 1, corresponding to a perfectly general layer. The first and second layers have transfer specificities of roughly 5, and are general (within error) at all widths. The fourth layer, on the other hand, has a transfer specificity of roughly $10^5$, and is highly specific at all widths. Whereas those layers' transfer specificities do not change significantly with width, the third layer becomes increasingly general as the width increases. Its transfer specificity decreases by roughly a factor of 4 from roughly 64 at width 8 to 18 at width 16. In Figure 3(c), by comparison, its SVCCA specificity drops from roughly 5% at width 8 to 2% at width 16. Thus the transfer specificity metric agrees with the main results of the SVCCA specificity: at all four widths, the first two layers are general, and the fourth is very specific; the third layer is specific, albeit much less so than the fourth, and becomes more general as the width increases.

Returning to Figure 5(a-d), recall that the remaining control groups also contain information about network structure. Any difference between the two selffer groups (the two blue series) indicates fragile co-adaptation. We note possible fragile co-adaptation at a width of 8, especially at $n = 3$. Future work should try measuring co-adaptation using the SVCCA, perhaps by measuring the similarities of different layers within the same network, as done by Raghu et al. [14]. Finally, any significant difference between the two retrained groups (the two dashed series) was meant to check if retraining transferred layers boosted recipient performance; however, this was not seen in any of our cases.

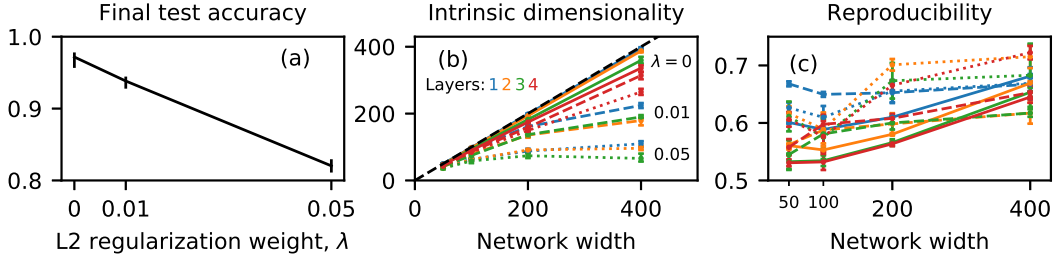

Figure 6: Test accuracies, intrinsic dimensionalities and reproducibilities of networks trained on the MNIST dataset for various L2 regularization weights $\lambda$ and network widths. The error bars in (a) and (b) show maxima and minima; those in (c) show the estimated standard error.

Overall, the transfer specificity used by Yosinski et al. [19] shows good agreement with the SVCCA specificity we defined. We note however that the SVCCA specificity is much faster to compute. Both methods require the training of the original set of networks without transfer learning, which took about 2 hours per network using our methodology and hardware. We could then compute all the SVCCA specificities for this work in roughly 15 minutes. On the other hand, Figure 5 required hundreds of extra hours to compute, and only considers four widths and two $x'$ values. That method would be prohibitively expensive for measuring generality in any continuously-parametrized problem.

### 3.4 Intrinsic dimensionality and reproducibility on MNIST

We also applied our metrics to the same networks trained instead on the MNIST dataset [9], and with ReLU activation functions rather than tanh. The networks were trained to minimize the classification cross entropy plus an L2 regularization term with weight $\lambda$. We used widths of 50, 100, 200, and 400; $\lambda$ values of 0, 0.01, and 0.05; and four random seeds per combination. Li et al. [11] measured the intrinsic dimensionalities of such networks, and found them to be vastly overparametrized.

Figure 6(a) shows, for each $\lambda$, the range of test accuracies over all widths after training on 2000 batches of 100 images. Figures 6(b) and 6(c) show the intrinsic dimensionalities and reproducibilities, respectively, by width, layer number, and $\lambda$. Without regularization (i.e. with $\lambda = 0$), the intrinsic dimensionalities of all four layers are nearly equal to their widths. This apparent contradiction to Li et al. [11] arises because their method is itself strongly regularizing. As we increase $\lambda$, the intrinsic dimensionalities decrease more rapidly than performance, which is consistent with the results of Li et al. [11]. We find low reproducibility in all experiments, even though the accuracies and intrinsic dimensionalities are quite consistent across seeds. This suggests that, at fixed width and regularization strength, although the networks consistently converge to representations of the same dimension, and although they exhibit comparable accuracy, the details of the learned representations vary significantly across seeds. In other words, the optimal representations in this experiment are non-unique. Since our metrics can be computed efficiently, future work should explore how this conclusion evolves during training. This experiment illustrates how our first two metrics, developed for DENNs, can also be applied more broadly. Our metric of specificity is based on a continuously-changing task; extending this to MNIST could be done, for instance, by varying the relative sampling of the target classes.

## 4    Conclusion

In this paper, we presented a method for measuring layer generality over a continuously-parametrized set of problems using the SVCCA. Using this method, we studied the generality of layers in DENNs over a parametrized family of BVPs. We found that the first layer is general; the second is somewhat less so; the third is general in wide networks but specific in narrow ones; and the fourth is specific for widths up to 192. We visualized the general components identified in the first layers and interpreted them as generalized coordinates capturing features of interest in the input domain. We validated our method against the transfer learning protocol of Yosinski et al. [19]. The methods show good agreement, but our method is much faster, especially on continuously-parametrized problems. Finally, we contrasted our intrinsic dimensionality with that used by Li et al. [11]. The two are distinct but complimentary, and produce consistent results for networks trained on the MNIST dataset [9].

**Acknowledgements**

MM gratefully acknowledges funding from the Ontario Graduate Scholarship (OGS). FZQ gratefully acknowledges funding from the Natural Sciences and Engineering Research Council (NSERC) in the form of Discovery Grant 2015-04533. HWdH gratefully acknowledges funding from the Natural Sciences and Engineering Research Council (NSERC) in the form of Discovery Grant 2014-06091.

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
