[Supplementary Material]

# Neural Networks Trained to Solve Differential Equations Learn General Representations: Supplemental Material

**Martin Magill**
U. of Ontario Inst. of Tech.
martin.magill1@uoit.net

**Faisal Z. Qureshi**
U. of Ontario Inst. of Tech.
faisal.qureshi@uoit.ca

**Hendrick W. de Haan**
U. of Ontario Inst. of Tech.
hendrick.dehaan@uoit.ca

## A   Additional implementation details

We used the $\tanh$ activation function for the DENNs in our work, and chose $\Omega = [-1, 1] \times [-1, 1]$ so that the input neurons had the same range as the hidden neurons. We used the $\tanh$ activation function as it outperformed the sigmoid, which was used in many previous works on DENNs (e.g. [2, 5, 6]). This is consistent with the general guidelines on efficient backpropagation offered by LeCun et al. [7]. LeCun et al. [7] also propose a rescaling of the $\tanh$ activation function that might improve performance when used with correspondingly rescaled inputs. Implementing this function using the python interface of TensorFlow [1] did not improve performance significantly, although a lower-level implementation might do better. Since this work was not performance-oriented, this is relegated to future work.

Piecewise-linear activations functions like ReLU were found to be incompatible with DENNs; the PDEs are defined in terms of the network's derivatives with respect to its inputs, and piecewise-linear activations functions produce solutions that are locally flat. Although such activation functions could still be used approximate the solution functions in theory (since Sonoda and Murata [10] showed that they still lead to universal approximation theorems), any function represented by them has no higher derivatives at any point in the domain, and so cannot learn by backpropagation from the loss functions used with DENNs.

Following Sirignano and Spiliopoulos [9], the loss function defined as

$$\mathcal{L}(x, y) = \left(\nabla^2 u - s\right)^2 \left(1 - I_{\delta\Omega}\right) + \eta u^2 I_{\delta\Omega} \tag{1}$$

where

$$I_{\delta\Omega}(x, y) = \begin{cases} 1, & (x, y) \in \partial\Omega \\ 0, & (x, y) \notin \partial\Omega \end{cases} \tag{2}$$

is the indicator function for the boundary of the domain. We chose $\eta = 1$, assigning equal weight to the PDE and loss terms. Since only one term is non-zero for any given point $(x, y)$, the relative importance of the PDE and BC are therefore controlled directly by the relative sampling of the interior and boundary of the domain.

In defining the loss function, the $L_2$ norm was consistently found to lead to better training performance than the $L_1$ norm. This was not clear *a priori*, as the problem studied here is essentially one of approximating a specific function, rather than learning from a statistical process. In this case, then, one might expect the $L_1$ norm to converge to sharper minima in the loss landscape, in much the same way that $L_1$ regularization encourages sparsification more readily than $L_2$ regularization. Future work should explore why training seemed to be less efficient with this loss norm.

During training, batches of training points were randomly sampled from the domain. Specifically, $10^4$ points were randomly drawn in the interior of $\Omega$, and then $10^4$ more points were drawn on *each* of

Figure 1: Final testing losses after training of all the networks used in the SVCCA-based generality measurements. Error bars show maxima and minima.

the four edges of the domain. Thus, although the loss function assigned equal weight on the PDE and BC terms, the data sampling favoured the boundaries significantly. The size of the training set was selected to optimally utilize the available GPU resources (NVIDIA GTX 1080). Since resampling the data set was computationally expensive, it was only changed every 100 training epochs.

The loss function was evaluated over a testing set of points, which was randomly generated in the same manner as the training set, but with ten times more points from each region of the domain (for a total of $5 \times 10^5$ points). This size was deemed to be more than large enough to fully resolve all features of the problem. As such, the testing set was only generated once for each experiment. The loss was computed over the testing set every 1000 epochs. Training proceeded until the testing loss failed to improve after five consecutive evaluations. This was found to reliably produce thoroughly-converged solutions in early tests, although the number of epochs before convergence varied significantly across different random initializations for the same experiments. As discussed below, this training protocol may have encountered issues for very wide networks.

Weights were randomly initialized according to the Tensorflow implementation of the Glorot uniform initializer [1, 3]. Optimization was conducted using the default TensorFlow implementation of the Adam optimizer [1, 4].

Because networks were fast to train (taking at most a few hours to converge), many instances of training (starting from different random seeds) were conducted for each experiment. In order to reduce the chance of artifacts arising from random seed correlations, the seeds were set according to the formula

```
seed = int(str(nxp+1) +
           "%02d"%(seed_core) +
           str(np.abs(nr)+1) +
           str(n_layers) +
           "%03d"%(neurons_per_layer))
```

where `nxp` indicates the number of increments of $0.1$ by which the source has been translated in $x'$, `nr` indicates number of times the effective width of the source was increased by a factor of 2 from $r = 0.1$, `n_layers` is the number of hidden layers in the network and `neurons_per_layer` is the number of neurons in each hidden layer. Finally, `seed_core` is a number use to distinguish between different repetitions of the same experiment. Note that several of these parameters were not varied in the current work, but this convention was selected for compatibility with future work.

Figure 1 shows the final testing losses achieved by all the networks trained for the SVCCA-based generality measurements. In other words, these are the losses for all networks trained for this work, except those trained using transfer learning. Performance improved with width until a width of 24. Wider networks achieved somewhat worse performance. We observed very wide networks during

Figure 2: Training statistics for all the networks used in the SVCCA-based generality measurements. Error bars show standard deviations.

training and noticed that they made incremental (in the third decimal place) improvements in testing loss for many testing periods before converging. We believe this to be due to very broad, flat regions near the minima of the loss landscapes of these networks. As discussed in the text, our training protocol may have terminated for these networks before they attained the minima of these plateaus. As a result, this may have interfered with the networks discovering general representations in the second and third layers. Similarly, if this coincided with co-adapation among the second, third, and fourth layers, then some of the generality observed in the second and third layers at moderate widths may have been shared with the fourth layer in the under-converged very wide networks. In other words, the increase in the SVCCA specificities of the second and third layers at very large widths could be related to the slight decrease in the SVCCA specificity of the fourth layer in the same networks. Certainly, future work should explore this issue more carefully. At a practical level, a different training protocol than that used here might be beneficial for training very wide DENNs in performance-oriented settings.

Figure 2 sumarizes the runtime performance of all the networks except those trained with transfer learning. Despite converging in fewer epochs, very wide networks were slowest to train, as they took the longest to train per epoch. Very small networks took longer to train because they took many more epochs to converge. Again, these behaviours might be of interest to future performance-oriented work.

## B    Additional analysis details

We used the SVCCA code provided by Raghu et al. [8] on the Google github repository. Their implementation included various threshold values used to remove small values from the data, as these are expected to correspond to noise. Because of the nature of DENNs, training is conducted with an unlimited amount of training data and without any noise in the data. As such, we did not use these thresholding operations.

The error bars of reproducibility and specificity in Figures 3 and 6 of the main text were obtained by treating the distributions of each of $\rho_{\text{self}}^l, \rho_{\Delta x'=0}^l$, and $\rho_{\Delta x'=X}^l$ as uncorrelated samples and applying standard rules for the propagation of uncertainty. In reality, these values are in fact somewhat correlated, so the error bars should be taken only as approximate uncertainties. However, since the reproducibility varies quite smoothly with network width and the specificity agrees quite well with the validation tests, we deem the metrics to be sufficiently well-resolved for the current work, and properly accounting for error correlations is relegated to future work.

## C    Additional visualizations of first layers

Figure 3 shows the first 9 components obtained by SVCCA with five different pairs of networks' first layers. All these networks had widths of 192. The numbers on the left indicate which networks were compared: $x'_1$ and $x'_2$ are the respective $x'$ values on which they were trained, and $s_1$ and $s_2$ are their

Figure 3: Each row shows plots of the first nine canonical components found by applying the SVCCA between the first layer of a network trained on $x' = x_1'$ and the first layer of a network trained on a different random seed at $x' = x_2'$, as indexed to the left of the plots. The number above each plot shows the correlation between the layers in the direction corresponding to that component.

respective random seeds. Thus the first and last rows show self-SVCCAs, which are equivalent to singular value decompositions. The numbers above each component show the canonical correlation computed by the SVCCA for that component. The last row of Figure 3 contains the same components shown in Figure 5 of the main text.

In all five cases, the leading 9 components have the same general structure. All rows contain a pair of components that capture the same information as the original inputs $x$ and $y$:

1. In row 1, components 1 and 4, although component 4 is slightly distored by mixing with another component.

2. In row 2, components 1 and 2.

3. In row 3, components 1 and 2.

4. In row 4, components 1 and 4, although component 4 is slightly distorted by mixing with another component.

5. In row 5, components 1 and 2.

The remaining components highlight the 9 regions of interest in the domain, as discussed in the main text. For instance, the top-left corner is present in the following components:

1. In row 1, components 3 and 8.

2. In row 2, component 4.

3. In row 3, component 4.

4. In row 4, components 2 and 3.

5. In row 5, components 3 and 4.

Figure 4 shows the first 27 components and their correlations for the layer shown in the first row of Figure 3. We note two things here. First, as mentioned in the first text, the correlation values drop

Figure 4: Plots of the first 27 components of the first layer shown in the first row of Figure 3.

drastically after the ninth component. This was the basis for our use of the self-SVCCA as a measure of intrinsic dimensionality. Second, we note that the 11 components following the first 9 still seem to capture coherent features over the input domain. Indeed, they appear analogous to higher-frequency Fourier modes found in spectral analysis. In contrast, components 21 and higher of the remaining 192 components are quite incoherent.