[Reviews · NeurIPS 2018]

Reviewer 1



The authors introduce a technique to measure the generality (extend to which they transfer to another task) of hidden layers in neural networks. The authors do this by analyzing the singular vector canonical correlation (SVCCA), which makes use of the output of the hidden layers evaluated for different points in the network’s input domain. The authors apply their technique on neural networks that are trained to solved differential equations. The results show the first two layers of a NN generalize, the third layer only generalizes depending on the width of the network and the last layer doesn’t generalize. The paper is a novel application of SVCCA (introduced at NIPS ’17 by Raghu et al) to the problem of measuring generality/transferability of different layers in a NN. The method is inspired by Yosinski et al’s definition of generality, but has a favorable computational cost compared to the original method. The original SVCCA paper used the metric to introduce freeze training (freeze early layers in the network after they become stable during training), measure learning dynamics, and compare two models trained on CIFAR with different initialisations. The submitted paper repeats some of those experiments in the context of DENNs and has a more specific focus on transfer learning. The paper is well written with clear and extensive experiments. My main concern is the significance of the findings, given that many were already hinted at/explored in the appendix of the original SVCCA paper. However given the depth of the experiments in the proposed paper I think it would be a worthy contribution to the conference. I would be interested to know if the findings on generality of different layers extends from the differential equation domain to image classification, e.g. it would be interesting to see results similar to figure 3 on the CIFAR dataset that SVCCA uses. ------------------------ I would like to thank the authors for the extensive feedback. Based on the results for the MNIST dataset I have decided to increase my score.

Reviewer 2



This paper looks at using neural networks on a family of differential equations (Poisson equations) to study the transferability of a layer of a neural network using SVCCA, a technique that can determine similarity between pairs of layers. This paper had many interesting original definitions, and I liked the way that intrinsic dimensionality (which would be an interesting thing to compare to the ICLR paper: https://arxiv.org/abs/1804.08838), reproducibility and specificity were designed. Comparing two different methods for transferability is also a good experimental validation of the method. I did find some parts of the paper hard to follow, in particular, I was confused by the "block diagonal" structure referred to in Figure 1. While I think I was able to follow Figure 2 (which was based off of Figure 1), I think both Figures could do with expanded captions. I also think the definitions of reproducibility, specificity, etc on page 5 could sit in their own lines with the defining equation numbered. Having it written inline in free flowing text makes it hard to keep track of all the definitions. My main concern with this paper is that it would have been very nice to see an implementation of at least the computationally more efficient method on a different task and/or dataset. E.g. after comparing to the transferability paper, one could try to transfer between e.g. MNIST and SVHN as an additional validation of the method. While I like the motivation and the definitions, with only the differential equation application, it might not be enough for acceptance just yet. UPDATE: Having read the author response (particularly the MNIST experiments and comparisons with Li et al) I am updating my score to accept.

Reviewer 3



SUMMARY The paper presents a series of experiments in which singular vector canonical correlation analysis (SVCCA) is used to measure the similarity between neural network representations learned from different random initializations and from slightly different problems. In most prior work, the generalizability of neural network representations is measured through transfer learning experiments, which can be computationally expensive; the key insight of this paper is that SVCCA provides an alternative way to measure feature generalizability that tends to give similar conclusions as transfer learning, but which is orders of magnitude faster to compute. The authors use SVCCA to perform extensive experiments measuring the degree to which differential equation neural networks (DENN) vary with different random seeds, and vary across related problem instances. Experiments with SVCCA suggest that lower network layers are more general, while higher layers are more specific to the particular problem instance on which the network was trained. Lower layers are more reproducible, giving rise to representations that are more stable under retraining, and across all layers reproducibility grows as network width increases. A subset of these experiments are reproduced with transfer learning, and give similar conclusions but take orders of magnitude longer to run. PROS - The paper demonstrates that SVCCA can be a computationally inexpensive way to study neural network representations - Experiments with SVCCA suggest a number of interesting features of neural networks - The paper is extremely well-written and clearly presented CONS - The motivation for focusing on DENNs is unclear - I don’t know how much the conclusions drawn from experiments on DENNs would generalize to other more commonly-used applications of neural networks. WHY FOCUS ON DENNS? The paper motivates the use of DENNs with two major reasons: (Lines 61-68): First, that DENNs are a powerful method for solving differential equations, and second that they “offer the opportunity to study the behavior of neural networks in a well-understood context”. I agree that these are both interesting reasons for studying DENNs; however as presented the bulk of the paper has very little to do with DENNs at all, and instead focuses on the degree to which neural network representations vary when trained in different conditions. The questions that the paper raises around the intrinsic dimensionality, reproducibility, and specificity of learned representations are very interesting and of extremely broad applicability; however given the far-reaching implications of these experimental questions, I’m surprised that the authors chose to focus on the relatively niche area of DENNs rather than more well-studied applications like image recognition, speech recognition, language modeling, or machine translation. HOW MUCH DO CONCLUSIONS GENERALIZE? The paper’s experiments suggest a number of intriguing conclusions about learned representations; however due to the focus on DENNs I’m unsure to what extent these conclusions would generalize to more standard applications of neural networks, especially since there are some quirks of DENNs that are quite different from many other types of neural networks. First, the problem setup and loss for DENNs are not very representative of typical applications of neural networks; the network inputs the (x, y) coordinates of a point the plane, and outputs a scalar value; the loss function (Supplementary Equation 1) incorporates a squared Euclidean distance between the second derivative of the network’s output and a target function s. This problem setup and loss function are extremely different from most applications of neural networks, which typically input some natural data (images, text, speech, etc) and are trained with cross-entropy or regression loss functions. Second, all networks studied are extremely small: they are four-layer fully-connected layers, with between 8 and 192 hidden units per layer. This means that the largest networks studied have on the order of 150K learnable parameters; in comparison a ResNet-50 on ImageNet has about 25M learnable parameters. Will the conclusions of the paper generalize to larger networks? Third, all networks use tanh nonlinearities; ReLU nonlinearites are much more common for most problems. The authors do provide a convincing explanation for why tanh is preferred over ReLU for DENNs (Supplementary Lines 10-16), though I still wonder whether the results would apply to ReLU networks. OVERALL On the whole I enjoyed reading the paper. It was extremely well-written, and used well-designed experiments and metrics to draw several interesting conclusions about neural network representations, demonstrating that SVCCA can be a valuable tool in understanding neural networks. However, due to the exclusive focus on DENNs I am unsure how much the conclusions will apply to other more widely-used types of neural networks; experiments on standard tasks (image classification, language modeling, etc) would have significantly strengthened the paper. Despite this shortcoming, overall I lean toward acceptance. FINAL SCORE I appreciate the additional experiments presented on ReLU networks on MNIST, and I hope these can be included in the camera-ready version of the paper.